# Comparison of Three Video Laryngoscopes and Direct Laryngoscopy for Emergency Endotracheal Intubation While Wearing PPE-AGP: A Randomized, Crossover, Simulation Trial

**DOI:** 10.3390/healthcare11060884

**Published:** 2023-03-18

**Authors:** Przemysław Kluj, Anna Fedorczak, Michał Fedorczak, Tomasz Gaszyński, Cezary Kułak, Mikołaj Wasilewski, Mateusz Znyk, Maria Bartczak, Paweł Ratajczyk

**Affiliations:** 1Department of Anesthesiology and Intensive Care, Medical University of Lodz, 90-549 Lodz, Poland; 2Department of Pediatrics, Nephrology and Immunology, Medical University of Lodz, 93-338 Lodz, Poland; 3Department of Endocrinology and Metabolic Diseases, Polish Mother’s Memorial Hospital-Research Institute, 93-338 Lodz, Poland; 4Medical Simulation Center, Medical University of Lodz, 92-213 Lodz, Polandmateusz.znyk@umed.lodz.pl (M.Z.);; 5Clinic of Anesthesiology and Intensive Therapy, Medical University of Lodz, 92-213 Lodz, Poland

**Keywords:** endotracheal intubation, paramedic, direct laryngoscopy, video laryngoscopy, personal protective equipment

## Abstract

Background: The COVID-19 pandemic has necessitated changes in the safety protocols of endotracheal intubation at every level of care. This study aimed to compare the first-pass success rates (FPS) and intubation times (IT) of three video laryngoscopes (VL) and direct laryngoscopy (DL) for simulated COVID-19 patient emergency intubation (EI). Methods: The study was a prospective, randomized, crossover trial. Fifty-three active paramedics performed endotracheal intubation with the I-view^TM^ VL, UESCOPE^®^ VL, ProVu^®^ VL and Macintosh direct laryngoscope (MAC) wearing personal protective equipment for aerosol-generating procedures (PPE-AGP) on a manikin with normal airway conditions. Results: The longest IT was noted when the UESCOPE^®^ (29.4 s) and ProVu^®^ (27.7 s) VL were used. The median IT for I-view was 17.4 s and for MAC DL 17.9 s. The FPS rates were 88.6%, 81.1%, 83.0% and 84.9%, respectively, for I-view, ProVu^®^, UESCOPE^®^ and MAC DL. The difficulty of EI attempts showed a statistically significant difference between UESCOPE^®^ and ProVu^®^. Conclusions: The intubation times performed by paramedics in PPE-AGP using UESCOPE^®^ and ProVu^®^ were significantly longer than those with the I-view and Macintosh laryngoscopes. The use of VL by prehospital providers in PPE did not result in more effective EI than the use of a Macintosh laryngoscope.

## 1. Introduction

The COVID-19 pandemic has created many new challenges for healthcare professionals worldwide. It was necessary to modify many of the previously used standards of care in a short time. Many modifications concerned procedures performed on airways. Paramedics treating patients suspected of having or infected with SARS-CoV-2 in prehospital care were often forced to make difficult decisions regarding airway management in respiratory failure.

The airway management in patients with COVID-19 requires medical personnel to wear appropriate personal protective equipment (PPE) [1,2,3,4], which, in the case of operators with limited experience, may cause additional complications in this procedure [5]. Infection of medical personnel is possible through direct contact with an infected patient but the risk is especially high during airway management [3,4,5]. Growing data suggest that healthcare providers are being infected with SARS-CoV-2 following tracheal intubation of COVID-19 patients [6,7]; therefore, the safety of the procedure is particularly important.

This study was designed to compare intubation time (IT) and first-pass success (FPS) among paramedics using personal protective equipment for aerosol-generating procedures (PPE-AGP) with video laryngoscopes (VL) and direct laryngoscopy (DL) in a simulated airway emergency. It has to be emphasized that we did not evaluate the impact of PPE-AGP on intubation parameters.

## 2. Materials and Methods

This study was designed as a prospective, randomized, crossover, simulation-based trial. Manuscript and study preparation followed CONSORT guidelines. Fifty-three active paramedics working as a part of the National Emergency Medical Services System performed endotracheal intubation with a MAC DL and three VL: I-view^TM^ (Intersurgical^®^, Wokingham, Berkshire, UK), ProVu^®^ (Flexicare Medical Ltd., Cynon Business Park, Mountain Ash, UK), UESCOPE^®^ (UE Medical Devices, Newton, MA, USA) (Figure 1) with PPE-AGP. The PPE used included the disposable medical protective clothing, the characteristics of which protect against high concentrations of organic and inorganic chemical particles and those with a diameter of fewer than one µm (Yunxiao Reton Outdoor Safety Sports Products Co., Ltd., Zhangzhou City, Fuijan, China). The eyeballs were protected with face shield (Cover One^®^, Tech Design, Wroclaw, Poland), and a disposable FFP2 mask with a filter was used to protect the subjects’ airways (Zhejiang Mashang Technology Co., Lingxi Town, Can Wenzhou Zhejiang, China). One pair of nitrile gloves was also used for hand protection (Figure 2).

We assessed the time to intubate and FPS rate among prehospital healthcare professionals. Instead of Cormack–Lehane classification, a short questionnaire consisting of four options to select (easy, complicated, difficult and very difficult) was used to examine participants’ subjective experiences of different intubation techniques wearing PPE.

All paramedics participated in a 30 min training course before starting the study. At the end of the main training, study participants had 15 min to familiarize themselves individually with all laryngoscopes under normal airway conditions on Laerdal Airway Management Trainer placed on the table in a neutral position. In the study, each participant performed endotracheal intubations using four laryngoscopes in a single airway scenario. Participants were randomized to start with one of four devices. In each scenario, participants had up to two attempts to intubate with each laryngoscope. It has to be emphasized that only the first intubation attempt parameters were assessed.

This study was conducted in two facilities from 2 February to 29 December 2022. The first one was the Medical Simulation Center of the Medical University of Lodz, and the second one was Emergency Medical Services School of Voivoid Rescue Station in Lodz City. Manikin used for the study was Laerdal Airway Management Trainer (Laerdal^®^, Wappingers Falls, NY, USA).

### 2.1. Direct Laryngoscopy

A standard MAC size 3 blade was used for DL along with a 7.5 mm internal diameter cuffed tracheal tube (Zarys International Group, Zabrze, Poland) for all intubation attempts. Time to intubation was defined as starting when the blade of laryngoscope was inserted between the teeth and timing finished at the first ventilation of the lung. Each time was measured using a stopwatch by the same instructor.

### 2.2. I-View^TM^ Video Laryngoscope

I-view^™^ is a single-use and fully disposable VL, providing the option of video laryngoscopy. Incorporating a one-size MAC blade makes the I-view insertion technique more familiar. Integrated with a handle, simple LCD screen is ready to use in seconds after removing from the packaging and provides an optimal view in various light conditions. The time and effectiveness measurements were the same as mentioned in the direct laryngoscopy section (including tracheal tube size).

### 2.3. ProVu^®^ Video Laryngoscope

A single-use blade, disposable ProVu^®^ VL with a mid-angulated MAC size 3 blade was used for the study. This VL is mainly equipped with an optimized focal camera, which offers a 60° field of view for an improved view of the glottic area, a 70° Display Tilt Screen, bright white LED light, internally mounted Anti-Fog design, and a quick start option for emergency intubation. The time and effectiveness measurements were the same as mentioned in the direct laryngoscopy section (including tracheal tube size).

### 2.4. UESCOPE^®^ Video Laryngoscope

A single-use, mid-angulated blade system D3 of VL460 UESCOPE^®^ VL was used in the current study. This type of laryngoscope offers a touchscreen monitor with high-definition video recording designed as a new standard for prehospital airway practice. Additionally, VL460 is equipped with 110° tilt and 270° rotate display. The time and effectiveness measurements were the same as mentioned in the direct laryngoscopy section (including tracheal tube size).

### 2.5. Statistical Methods

The statistical analysis of collected data was performed using STATISTICA ver. 13.3 software (Statsoft, Poland). Shapiro–Wilk test was used for the assessment of normal distribution. Nonparametric ANOVA Friedman with post hoc tests and Q Cochrane tests for repeated measures and multiple comparisons were used for statistical analysis. A *p*-value < 0.05 was considered statistically significant.

For this study, the sample size was based on G*Power 3.1 statistical tool, using a 2-tailed *t*-test. A minimum of 40 participants were necessary to achieve a Cohen d = 0.8 (alpha error = 0.05, and power = 0.95). To provide a safety margin in case of missing data or nonparticipation, we increased the minimum size of the study group to 53 (*n* = 53) participants.

### 2.6. Ethical Approval

This research was a manikin study with volunteers and approval by the institute’s ethics committee was not required. However, we received ethical approval for this study from The Medical University of Lodz Ethics Board (Protocol Signature: RNN/06/20/KE).

## 3. Results

The results show the data obtained for endotracheal intubation performed using three VL and MAC laryngoscope when using PPE-AGP. We did not perform any intubation measurements without wearing PPE-AGP. The study strictly aimed to compare four different types of laryngoscopes in a simulated COVID-19 airway management scenario.

Fifty-three (*n* = 53) paramedics were included in the study. Forty-one (78%) were men and twelve (22%) were women. Their mean age was 31.9 ± 9.5 years. Overall, participants performed 212 manikin intubations. All participants managed to intubate the manikin, using three VL and MAC laryngoscope, in the study with PPE within the 11-month study period. The average professional experience of the paramedics in instrumental airway management was 9 years. Both the order of participants and intubation methods were randomized using the Research Randomizer program. Each participant performed intubation in one scenario setting. There were no drop-outs (Figure 3).

The primary outcome was the FPS time of the emergency intubation (EI) attempt performed by the paramedics. There was a statistically significant difference in intubation time by different VL and MAC laryngoscopes (*p* = 0.00041). The longest IT was noted when the UESCOPE^®^ was used (29.4 s). The median intubation time for ProVu^®^ was 27.7 s, for I-view 17.4 s and for MAC 17.9 s. The post hoc test for ANOVA Friedman revealed statistically significant differences in IT between UESCOPE^®^ and ProVu^®^ vs. I-view and MAC (*p* < 0.001) In the case of I-view vs. MAC and UESCOPE^®^ vs. ProVu^®^ there was no significant difference in the IT. Detailed information is presented in Table 1 and Figure 4.

The secondary outcome was the effectiveness of intubation attempts. The FPS success rate was 88.6% for I-view, 81.1% for ProVu^®^, 83.0% for UESCOPE^®^ and 84.9% for MAC DL. There was no significant difference in the success rate of the FPS attempt between devices. Detailed information is presented in Table 2 and Figure 5.

As with the other measures, we evaluated the degree of difficulty of the intubation attempts. There was a statistically significant difference in the difficulty level between the VL (*p* = 0.00143). Post hoc analysis showed that there is a statistically significant difference in the difficulty level between UESCOPE^®^ and ProVu^®^. Moreover, 45.2% of intubation attempts performed by UESCOPE^®^ and 34.0% by MAC were difficult, and only 11.3% and 13.2% were easy. Comparatively, most of the intubations performed by I-view and ProVu^®^ were easy (45.2% and 39.6%), complicated (37.9% and 41.6%) or difficult (16.9% and 18.8%). None of the intubation attempts were considered very difficult. Detailed information is presented in Table 3 and Figure 6.

## 4. Discussion

The COVID-19 pandemic has brought all intubation techniques into further focus to secure the safety of airway management at every level of care. The present study demonstrates the advantages and disadvantages of using a MAC laryngoscope and three VL when it is necessary to secure the airway in an environment exposed to the formation of highly infectious aerosols. Additionally, we assessed the degree of difficulty of intubation attempts on a standard airway manikin.

Widespread use of VL in hospital practice requires special evaluation and attention before placing this airway management method into out of hospital care. The negative impact of different levels of PPE on various airway management skills among healthcare providers is indicated in numerous publications [5,8,9,10]. Therefore, in our study we did not assess the impact of PPE-AGP on the intubation parameters. We are considering conducting such a study in the future.

EI has long been considered as the criterion standard for securing the airway in prehospital care. The experience of paramedics concerning intubation in DL is limited and has been questioned as the preferred method of advanced airway management [11,12]. Intubation performed in DL is one of several available options for an advanced airway management strategy. Alternatives to EI include supraglottic airway devices including the laryngeal mask airway, i-gel and laryngeal tube. On the other hand, growing data suggest a VL is a good option for paramedics [13]. Despite controversy over the idealized form of airway management, multiple observational studies have reported better outcomes for patients who received EI via Emergency Medical Services (EMS) Paramedics [14,15,16,17].

The use of a manikin does not equate to actual airway management techniques on patients in prehospital care. Manikin studies play an important role in airway management studies, but their benefits for clinical practice are still a matter of debate [18].

On the other hand, this study allows an important practical comparison of three types of VL wearing PPE while undertaking various intubation techniques. If using new types of VL among less experienced operators impedes the ability to perform tracheal intubation, then its use during high-risk procedures on real patients could be dangerous. Furthermore, there is continuing concern that wearing PPE will hinder the intubator’s performance, possibly owing to lack of experience, a high stress level, limited communication, restricted movement, impaired manual dexterity and restricted visibility. All these factors are clinically significant and may seriously affect patient safety [4,8,9].

Several recent studies have compared the efficacy of various intubating devices while the intubator was wearing PPE and have shown ambiguous results. Ludwin et al. reported that PPE reduces the effectiveness of EI. The use of DL for intubating patients with suspected/confirmed COVID-19 by an operator wearing level C PPE is associated with overall intubation time reduction and an increase in the intubation success rate compared with VL (93.6% vs. 92.3%, *p* = 0.66). However, reported findings suggest that MAC blade VL during endotracheal intubation with PPE may be an alternative to DL. Additionally VL can be helpful for less experienced personnel [13]. Gadek et al. reported that the McGrath MAC VL possesses statistically significant advantages in time (24.8 s vs. 34.0 s, *p* < 0.001) and effectiveness of EI over the MAC DL (100% vs. 83%, *p* = 0.002) when used by paramedics in PPE in a manikin-simulated study [19]. In another simulation study, Shin et al. compared Pentax-AWS (AWS) and DL while the intubator wore chemical, biological, radioactive and nuclear (CBRN) PPE. The authors found that AWS required less time to complete an EI than a DL (18.2 s vs. 26.4 s, *p* < 0.001) [20]. In a study that compared the EI using McGrath MAC VL and MAC DL in PPE among novice physicians, the authors reported a significantly prolonged time of successful EI when a standard laryngoscope was used (24.0 s vs. 14.0 s, *p* < 0.001). Additionally, the FPS rate was significantly higher in the VL group (94.3% vs. 58.6%, *p* < 0.001) [21]. In a recently published study, performed on 756 patients intubated within the Emergency Department by emergency physicians in PPE, Kim et al. demonstrated a significantly increased FPS rate when a C-MAC VL was used instead of a MAC DL (85.7% vs. 69.8%, *p* < 0.001) [22].

Conversely, in a manikin study, Yousif et al. demonstrated that the use of VL by prehospital providers in Level C PPE did not result in faster EI than the use of a MAC DL (29.9 s vs. 25.7 s, *p* < 0.001). The King Vision VL, in particular, performed at least as well as the MAC DL and was reported to be easier to use [23]. Sule et al. reported that median times were higher using VL than DL (86 s vs. 78 s, *p* = 0.159), but there was no significant difference between intubation with either DL or VL in subjects with and without Level C PPE [24]. In a study conducted by Goh et al., the authors compared the McGrath VL and a DL as used by specialized anesthetists wearing PPE-AGP and N95 masks on 28 patients undergoing elective surgery. In that study, the median IT for VL (61 s) and DL (41.5 s) did not differ significantly, and they found no statistically significant differences in the FPS rate [25].

The UESCOPE^®^ and ProVu^®^ are relatively new devices for airway management. To the best of the authors’ knowledge, no published data are available on the use and comparison of presented VL while wearing PPE-AGP, either in real-life medical conditions or with the use of medical simulation. Therefore, we selected them for our evaluation.

In our study, the FPS rate wearing PPE was relatively high but did not differ significantly among devices. When comparing the intubation times across all four groups in our study, we found that the IT when paramedics were using UESCOPE^®^ and ProVu^®^ was significantly prolonged compared to I-view and MAC DL.

According to our observation and specific comments provided by the paramedics, the prolongation of the EI time was associated with a necessity to use a profiled guide attached to UESCOPE^®^ and ProVu^®^, which is a common problem in less experienced operators. Additionally, paramedics claimed that it is difficult to maneuver the laryngoscope handle from a distance and use the guide at the same time. Our study is strengthened by the fact that all intubation attempts were conducted by paramedics who have been actively engaged in clinical practice within EMS Teams during the COVID-19 pandemic.

The thing that is not important from our point of view but has to be emphasized, is the overall EI time, which in our study achieved a better outcome while using the cheaper and the older laryngoscopes. Moreover, it is worth noting that our findings are highly relevant to the clinical situation and suggest that utilizing new types of VL wearing PPE in most cases does not complicate intubation performance but extends the time of the procedure.

The study we conducted has several limitations. The study is a pilot study on relatively newly introduced devices. Only three types of VL and DL were compared. A range of VLs are available in the marketplace, and they have considerable heterogeneity in blade and screen design (i.e., MAC-type blade vs. hyperangulated blade, HD screen vs. LCD screen). Because different devices could provide different findings, caution should be used when extending our results to other contexts. Moreover, the results obtained cannot be directly transferred to clinical practice due to the simulation nature of the study. An important factor is also the issue of the experience that individual study participants have in airway management. The study was limited to paramedics working in a prehospital environment and cannot be directly transferred to physicians and other medical personnel with different experiences in airway management. Therefore, our study results should be interpreted in the wide context of instrumental airway management performed by paramedics using different VL because the type and extent of PPE-AGP used in our study might also differ from those in other studies. Differences in performing effective intubation between the evaluated airway devices may be significant. In our study we compared devices with a MAC-like blade (one was a mid-angulated-type) with a mid-angulated-D3-blade-type system. This may be considered as a potential limitation. However, the use of a hyperangulated blade VL was described in several studies as more difficult than the use of devices with MAC-like blades. In the paper of Pieters et al. comparing seven airway VL, the authors revealed that the type of device is important for the time of successful intubation [26]. Better results were achieved when a MAC-like blade VL was used, compared to hyperangulated devices [26,27]. Based on this and referring to the obtained results, we assume that it may be more relevant for clinical practice to evaluate devices with a standard MAC-like blade.

Our findings provide an additional overview on EI performed by paramedics in PPE-AGP utilizing new types of VL. It seems that DL is a much more familiar method of EI for paramedics in the manikin scenario rather than an EI performed by UESCOPE^®^ and ProVu^®^ VL. The time of successful EI in our study was significantly prolonged for advanced types of VL, which is an interesting finding and obviously requires further evaluation with a larger group. It is possible that the DL was more familiar to the intubators than the VL because it was used more frequently during their clinical practice.

The prolongation of successful EI performance exceeding >10 s in a manikin scenario seems to be clinically important (taking into account the low stress level in comparison with real patient intubation), but as previously mentioned, it cannot be simply transferred to the prehospital practice. On the other hand, VL was recommended as the preferred intubation method during the COVID-19 pandemic to keep a meaningful distance between the patient’s and the operator’s airway to minimize the risk of aerosol exchange. Thus, even if the DL show better results, with respect to personal protection during procedures with real patients, the use of VL should be considered.

## 5. Conclusions

In conclusion, the use of video laryngoscopes by prehospital providers under PPE-AGP-wearing conditions in a randomized, crossover simulation trial did not result in more effective endotracheal intubation than using a standard Macintosh laryngoscope. The intubation time using the UESCOPE^®^ and ProVu^®^ video laryngoscopes was significantly prolonged compared to the I-view and Macintosh laryngoscopes. We support the approach that the intubation technique requires more practice to obtain better overall outcomes among paramedics, especially in circumstances where intubator performance is hindered by PPE-AGP.

## Figures and Tables

**Figure 1 healthcare-11-00884-f001:**
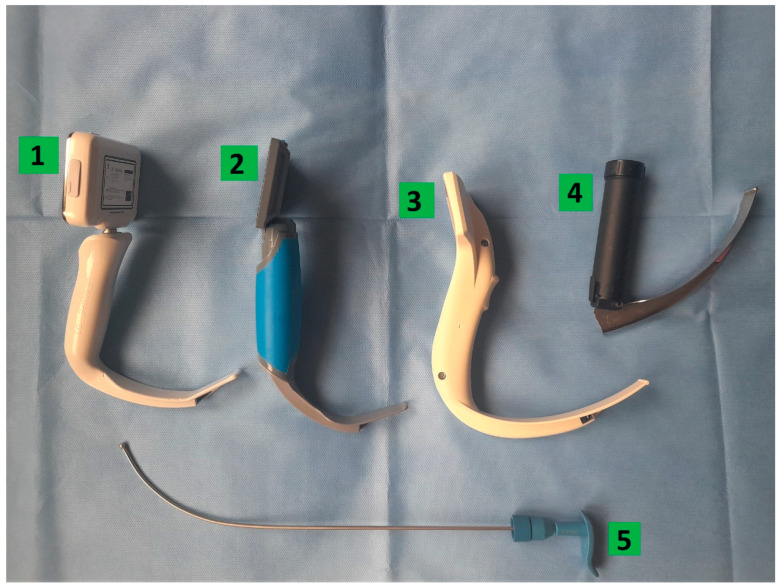
Laryngoscopes used in the study—**1**: UESCOPE^®^; **2**: ProVu^®^, **3**: I-view^TM^: **4**: Macintosh; **5:** profiled intubation guide.

**Figure 2 healthcare-11-00884-f002:**
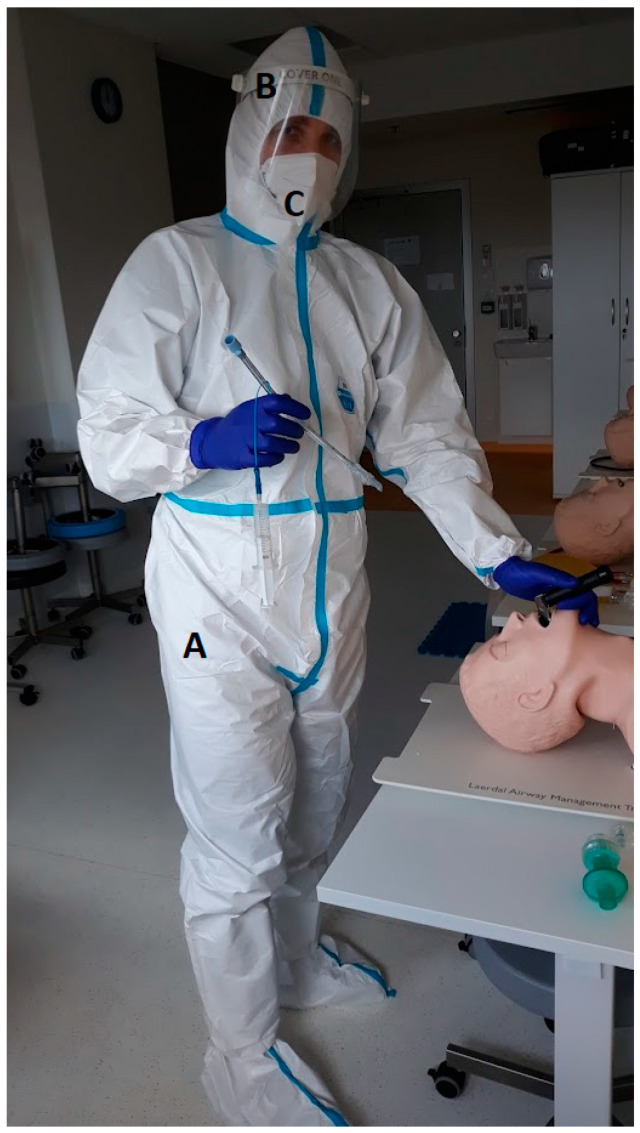
PPE-AGP used in the study—**A**: medical protective clothing; **B**: face shield; **C**: disposable FFP2 mask.

**Figure 3 healthcare-11-00884-f003:**
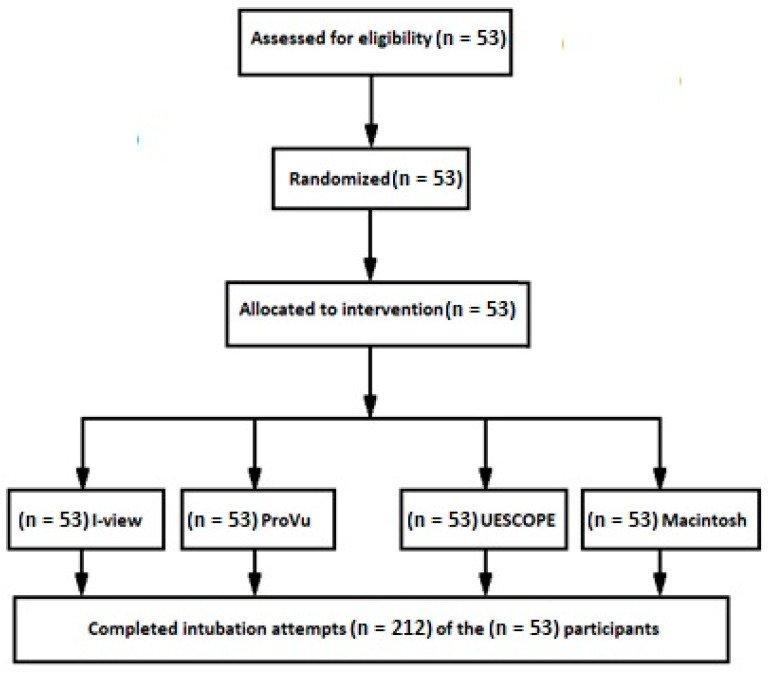
Randomization flow chart.

**Figure 4 healthcare-11-00884-f004:**
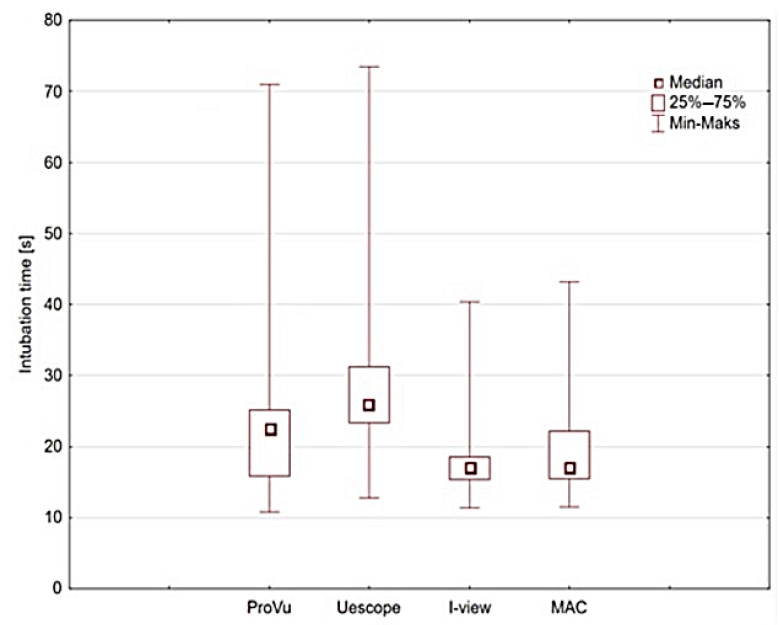
Median time to successful intubation by different types of video laryngoscopes and classic Macintosh laryngoscope (MAC).

**Figure 5 healthcare-11-00884-f005:**
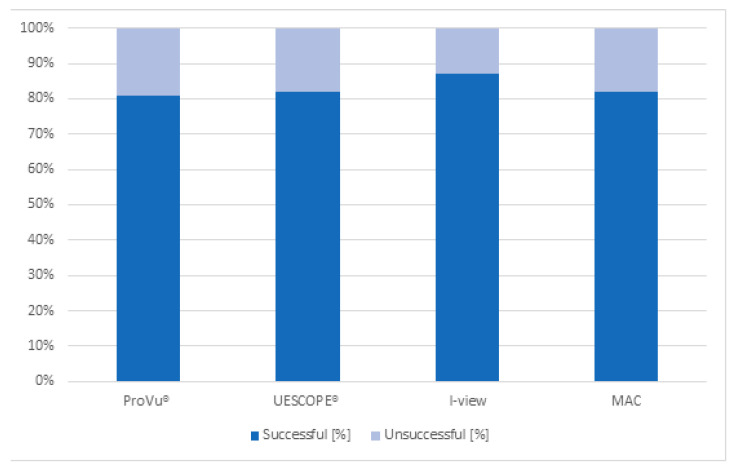
Number of successful and unsuccessful intubation attempts.

**Figure 6 healthcare-11-00884-f006:**
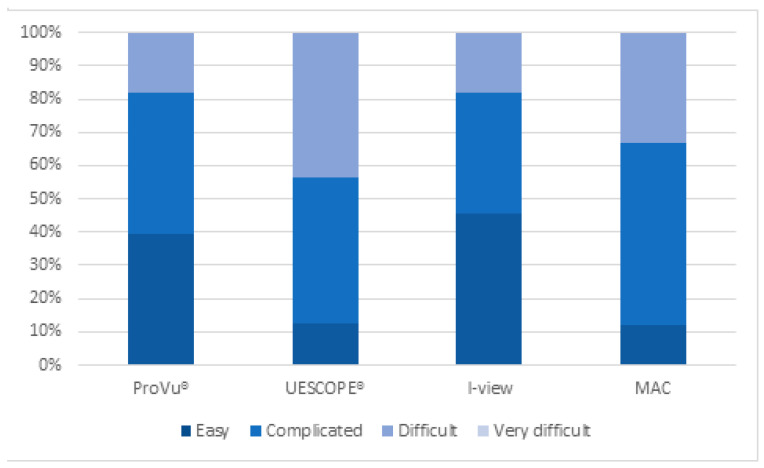
Difficulty level of each (video) laryngoscope.

**Table 1 healthcare-11-00884-t001:** Time characteristics for successful intubation (seconds).

Video Laryngoscope Type	Median	Mean	Minimum	Maximum	Lower Quartile	Upper Quartile	Standard Deviation
ProVu^®^	27.7	28.8	11.8	70.5	19.7	29.8	9.2
UESCOPE^®^	29.4	28.1	12.8	73.4	22.3	33.9	10.1
I-view	17.4	16.0	10.3	40.8	14.3	17.5	4.6
MAC	17.9	16.1	10.5	43.2	14.4	22.2	5.2

**Table 2 healthcare-11-00884-t002:** Number of successful and unsuccessful intubation attempts with each video laryngoscope.

Video Laryngoscope Type	Successful		Unsuccessful	
	Number	Percentage	Number	Percentage
ProVu^®^	43	81.1%	10	18.9%
UESCOPE^®^	44	83.0%	9	17.0%
I-view	47	88.6%	6	11.4%
MAC	45	84.9%	7	15.1%

**Table 3 healthcare-11-00884-t003:** The difficulty level of each (video) laryngoscope. Chi^2^ ANOVA (N = 32, df = 3) = 15.50870, *p* = 0.00143.

Video Laryngoscope Type	Difficulty Level							
	Easy		Complicated		Difficult		Very Difficult	
	Number	Percentage	Number	Percentage	Number	Percentage	Number	Percentage
ProVu^®^	21	39.6%	22	41.6%	10	18.8%	0	0%
UESCOPE^®^	7	13.2%	22	41.6%	24	45.2%	0	0%
I-view	24	45.2%	20	37.9%	9	16.9%	0	0%
MAC	6	11.3%	29	54.7%	18	34.0%	0	0%

## Data Availability

Data files available on request.

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
