# Peer review of "Comparison of Three Video Laryngoscopes and Direct Laryngoscopy for Emergency Endotracheal Intubation While Wearing PPE-AGP: A Randomized, Crossover, Simulation Trial"

_healthcare, 2023, doi:10.3390/healthcare11060884_

Round 1

Reviewer 1 Report

This is an interesting practical study comparing the intubation time and first pass success rate of 3 video laryngoscopes (VL) and a direct laryngoscope by 53 experienced pre-hospital paramedics in full PPE.

The results are of interest and reflect not all VLare the same with two of the scopes having a significant longer intubation time although success rates were similar. It is noted that DL in relatively experienced paramedics hands in this model is comparable in intubation time to the best if the VL (I-view).

It is a manikin study and is not directly related to the real life situation but the authors have explained the limitations and give a good overview of similar studies.

The study design is robust and it is well written.

Reviewer 2 Report

Thank you for the possibility to read this manuscript.

The authors compared direct laryngoscopy using a standard Macintosh laryngoscope withe three different video laryngoscopes using PPE. As there was no study group without PPE, the trial focuses on the differences in intubation success using the different laryngoscopes.

From my point of view the authors did a mistake in describing the shapes of the used blades, as on page 10 line 312 the define to compare only laryngoscopes with MAC-like blades. From my point of view this is in contrast to Fig. 1: While the blade No.3 seems comparable to the standard MAC-blade (No.4), No.1+2 (UESCOPE + PROVU) do not. As this corresponds with unsuccessful intubation attempts, it should be discussed in the manuscript.

Helpful references may be:

Kelly at al. Seeing is believing, BJA 2016; 117: i9-i13

Lafferty et al. Video laryngoscopy as a new standard of acre, BJA 2025; 115: 136-137

In the methods section is described that the participants had a 30 minutes training and  a 15 minutes familiarization time, but the experience level in using direct laryngoscopy is not mentioned.

A further topic is that video laryngoscopes have been recommended during the pandemic to keep a meaningful distance between the patient's and the operator's airway to minimize the risk of aerosol exchange. Thus, even if the direct laryngoscopy show better results, with respect to personal protection its use should be discussed as well.

All these facts should be addressed before publishing this manuscript.

Reviewer 3 Report

I thank the authors for giving me the opportunity to read this interesting paper. In order to improve the manuscript, I would like to make some comments.

The authors indicate in the abstract some terms with their acronyms in parentheses. However, they do not do so in the text. The meaning of each acronym should be indicated in the text the first time it appears, as was done in the abstract. There will be readers who will read the text without reading the abstract and should understand what the acronyms mean without having to look them up outside the text. E.g. PPE - line 43, IT, FPS, PPE-AGP - line 49, etc.

Figures 5 and 6 do not provide relevant information that improves on that provided by the corresponding tables 2 and 3.

In section 3.3 and Table 3 the authors present the difficulty level of each laryngoscope. I cannot find in the "material and methods" section the description of how the level of difficulty was measured. 

In line 70 they indicate that a short questionnaire was performed. What did it consist of? I do not see the data in the results. It should be adequately explained. 

Finally, I congratulate the authors for the work done.
